# Impact of Acid Suppression Therapy on Renal and Survival Outcomes in Patients with Chronic Kidney Disease: A Taiwanese Nationwide Cohort Study

**DOI:** 10.3390/jcm11195612

**Published:** 2022-09-23

**Authors:** Yi-Chun Chen, Yen-Chun Chen, Wen-Yen Chiou, Ben-Hui Yu

**Affiliations:** 1Division of Nephrology, Department of Internal Medicine, Dalin Tzu Chi Hospital, Buddhist Tzu Chi Medical Foundation, Chiayi 622, Taiwan; 2School of Medicine, Tzu Chi University, Hualien 970, Taiwan; 3Division of Hepato-Gastroenterology, Department of Internal Medicine, Dalin Tzu Chi Hospital, Buddhist Tzu Chi Medical Foundation, Chiayi 622, Taiwan; 4Department of Radiation Oncology, Dalin Tzu Chi Hospital, Buddhist Tzu Chi Medical Foundation, Chiayi 622, Taiwan

**Keywords:** histamine-2-receptor antagonist, proton pump inhibitor, dose-response relationship, CKD, ESRD, mortality

## Abstract

Histamine-2-receptor antagonist (H2RA) has shown beneficial effects on the kidney, heart, and sepsis in animal models and on the heart and COVID-19 infection in clinical studies. However, H2RAshave been used as a reference in most epidemiological studies examining the association of proton pump inhibitors (PPI) with outcomes. Therefore, we aimed to evaluate the effect of H2RA on renal and survival outcomes in chronic kidney disease (CKD) patients. We used a Taiwanese nationalhealth insurance database from 2001 to 2016 to screen 45,767 CKD patients for eligibility. We identified new users of PPI (*n* = 7121), H2RA (*n* = 48,609), and users of neither PPI nor H2RA (as controls) (*n* = 47,072) during follow-up, and finally created 1:1:1 propensityscore-matchedcohorts; each cohort contained 4361 patients. Participants were followed up after receivingacid-suppression agents or on the corresponding date until the occurrence of end-stage renal disease (ESRD) in the presence of competing mortality, death, or through the end of 2016. Compared toneither users, H2RAand PPI users demonstrated adjusted hazard ratios of 0.40 (95% confidence interval, 0.30–0.53) for ESRDand 0.64 (0.57–0.72) for death and 1.15 (0.91–1.45) for ESRD and 1.83 (1.65–2.03) for death, respectively. A dose-response relationship betweenH2RA use with ESRD and overall, cardiovascular, and non-cardiovascular mortality was detected. H2RA consistently provided renal and survival benefits on multivariable stratified analyses and multiple sensitivity analyses. In conclusion, dose-dependent H2RA use was associated with a reduced risk of ESRD and overall mortality in CKD patients, whereas PPI use was associated with an increased risk of overall mortality, not in a dose-dependent manner.

## 1. Introduction

Histamine-2-receptor (H2R) antagonist (H2RA) and proton pump inhibitor (PPI) are the main options for acid suppression therapy available at present, with PPI being more expensive, high-profile, and over-prescribed worldwide [1] than H2RA. Experimental data has shown a pathophysiological role of histamine signaling and H2R activation in response to histamine in the kidney, heart, and immune response [2,3,4,5]. H2Rs are found in mammalian and human cardiomyocytes, throughout the kidney and renal vessels, in the stomach and in neutrophils [2,3,4,5]. Histamine (via the H2R) mediates inflammation, which is central to renal diseases, reduces renal blood flow and creatinine and urea clearance at high doses, and modifies activated neutrophils, triggering oxidative burst and the release of reactive oxygen species; H2R activation stimulates renin release, promotes cardiac fibrosis and apoptosis, and plays a role in immune response [2,3,4,5]. Therefore, it seems biologically plausible that H2RA haspotential beneficial effects on the kidney, heart, and survival. However, accumulating epidemiological studies, which took H2RA as a reference, in the general population [6,7] and among Unites States veterans [8,9,10,11] reported increased risks for acute kidney injury (AKI) [6], chronic kidney disease (CKD) [6,8,9], end-stage renal disease (ESRD) [8,9], and mortality [10,11] in PPI users, despite a lack ofanimal models and firm biological bases to support plausible mechanisms of PPI-related renal harm and any impact on survival [9,12].

Animal interventional models of ischemic acute renal failure, pressure-induced heart failure, and sepsis and clinical studies of patients with heart failure and COVID-19 infection have provided evidence that H2RA could improve heart and renal functions, such as reduced levels of serum creatinine and blood urea nitrogen, and lower mortality [3,4,5,13,14,15,16,17,18]. However, there is a lack of clinical human evidence to address the impact of H2RA on the renal outcome. High plasma histamine levels have been found in CKD patients and are speculated to be fibrinogenic, which could contribute to cardiac fibrosis in isolated heart tissues of patients with heart failure [3,4,5]. This raises the question of whether H2RA could improve renal and survival outcomes in CKD patients.

CKD is associated with an increased incidence of acid peptic disease [19], which can be treated with antacids [20] containing ingredients such as aluminum, calcium, magnesium, or sodium bicarbonate, H2RAs, and PPIs. In Taiwan, antacids and H2RAs can beprescribed for patients having symptoms of an acid peptic disease without endoscopic examinations; however, PPIs, which are costly, are only reimbursed for patients who complete an endoscopic examination showing any type of acid peptic disease and prescribed only for 4 months since inspection date of each endoscopy. Given the H2RA and PPI impact on renal outcome, we implementedanew-user design, employed propensity score matching, and competing risk analysis toreappraise the renal and survival outcomes of CKD patients who were prescribed new H2RAs vs. new PPIs vs. neither substance as controls [21]. We analyzed long-term single-payer Taiwan’s National Health Insurance (NHI) claims data with more than 99% coverage. The dose-response relationship of acid suppressants with renal and survival outcomes was also evaluated.

## 2. Materials and Methods

### 2.1. Data Source

This was a retrospective cohort study using claims data from the 2005 Longitudinal Generation Tracking Database (LGTD2005), a de-identified database of 2 million individuals who were randomly sampled from all beneficiaries of Taiwan’s single-payer compulsory NHI program between 2000 and 2016. This study did not require informed consent and was exempted from full review by the Institutional Review Board of the Dalin Tzu Chi Hospital (B10603017, B10804003). Further details about the LGTD2005 have been described in our previous research [22]. In brief, the LGTD2005 does not differ from the NHI program and contains detailed medical information, except laboratory and lifestyle data. It adopts ICD-9/10-CMdiagnosis codes to define diseases andanatomical therapeutic chemical codes to capture drugs.

### 2.2. Study CKD Population

We included 405,767 patients who had a primary diagnosis of CKD between 1 January 2001 and 31 December 2016 from the outpatient and inpatient claims (Figure 1). The following patients were excluded: missing data for year and gender, less than 18 years old, ESRD onset prior to PPI or/and H2RA prescription, and who dropped out before CKD inception date. The resulting study population consisted of 137,031 CKD patients. Patientson first-time PPI prescriptions containing esomeprazole, lansoprazole, omeprazole, pantoprazole, or rabeprazole without H2RAs prescribed during follow-up were counted as new PPI users (*n* = 7121) and those on first-time H2RA prescriptions containingranitidine, cimetidine, and famotidine without PPIsprescribedduring follow-up were counted as new H2RA users (*n* = 48,609). Patients without the use of PPIsor H2RAs during follow-up were counted as nonusers (i.e., controls) (*n* = 47,072), which was also used in another study [21] of CKD patients investigating the association between PPI, H2RA, and renal and survival outcomes. Patients treated with both PPIsand H2RAs at any point in time during follow-up were excluded (*n* = 34,229). No ESRD occurrence was ascertained again between CKD inception date and first-time acid suppression therapy prescription. We considered the following covariates: preexisting comorbidities within one year before CKD inception date including hypertension, diabetes, coronary heart disease, chronic liver disease, and acid peptic diseases such asupper gastrointestinal bleeding, reflux esophagitis, and peptic ulcer disease [7], the number of medical visits within one year before CKD inception dateto minimize the detection bias [23,24] because of easy accessibility and availability of medical services in Taiwan and confounding effect of medical attentionas medical attention may explain some of the remaining risk elevation [23], and two confounding drugs non-steroidal anti-inflammatory drug (NSAID) and angiotensin-converting enzyme inhibitor/angiotensin receptor blocker (ACEI/ARB). To prevent immortal bias, we used the new-userdesign [22,25,26] with follow-up for each H2RA user beginning on the date of their first H2RA prescription and each qualified propensity-matched PPI user and control must have been alive at the time when H2RA commenced. Each new H2RA user was matched with one new PPI user (c-index, 0.68) with interval of less than 0.5 year when H2RA and PPI commenced and one control (c-index, 0.65) in the propensity score, which wascalculated by the logistic regression built on the matching variables including age, sex, comorbidities, the number of medical visits, and the same CKD year. The index date of the H2RA cohortwas the day of firstprescription of H2RA and that of the PPI and control cohorts was the corresponding matched day [22,26,27]. We did not use a time-dependent exposure design because the duration of a potential carry-over effect of H2RA or PPI in influencing renal and survival outcomes was unknown [14,22,26]. Thus, all patients were assigned to an exposure group basedon the drug that they were initially prescribed (H2RA or PPI).

### 2.3. Statistical Analysis

The three cohorts were followed from their index date to the occurrence of ESRD, death, or the end of 2016, whichever ensued first; the latter two were considered as censored observations. Nevertheless, death before reaching ESRD, which could lead to informative censoring, was considered a competing risk event in estimating the cumulative incidence and risk of ESRD [23,26]. ESRD was ascertained from the Registry for Catastrophic Illness Patient Database [22,23,26], a subset of the LGTD2005. All Taiwanese patients with ESRD requiring long-term dialysis can obtain a catastrophic illness certificate after a rigorous review by the NHI Administrationto be exempted from copayments for healthcare. Death was defined as withdrawal of the patients from the NHI program [25]. We compared baseline characteristics between the PPI, H2RA, and control cohorts using Chi-square and ANOVA tests for categorical and continuous variables, respectively. The modified Kaplan–Meier method and Gray’s method [28] were used to calculate and compare the cumulative incidences in data with competing risks. Meanwhile, we also analyzed Kaplan–Meier curve for ESRD-free survival in three study cohorts by log-rank test. After ensuring the assumption of proportional hazards by plotting the graph of the log (−log(survival)) versus the log of survival time, we applied the modified Cox proportional hazard model to examine the association of overall and individual acid suppressants with ESRD and Cox regression for mortality, with adjustment for all covariates listed in Table 1. To assess the dose-dependent association of acid suppressants with risks of ESRD, overall, cardiovascular, and non-cardiovascular mortality, we calculated each patient’s cumulative defined daily dose (cDDD) of acid suppressants according to the WHO’s recommendation [29] and divided the cDDDs into two levels by their median dose [30]. The cDDD of PPI was further subdivided into four levels (15, 30, 45, >45) based on its 30 median cDDD to elucidate detailed associations of PPI use with study outcomes. Meanwhile, the cDDD of H2RA was further subdivided into five levels (5, 10, 15, 20, >20) to elucidate detailed associations of H2RA use with study outcomes. Cardiovascular mortality was defined as death attributable to any cardiovascular event including heart, brain, or blood vessels. We also explored the relationship between the frequency of acid-suppressant prescriptions and study outcomes. To assess the reliability of our main findings, we conducted sensitivity analyses. First, multivariate stratified analyses were conducted for different subgroups. Second, we excluded CKD patients who died or developed ESRD within 30, 60, and 90days after the index date to reappraise the risk of study outcomes. Third, we implemented four models. Model 1 was to add anti-platelet and anti-lipid drugs into the original regression model shown in Table 2. Model 2 was to exclude the same CKD year from matching variables, redefine comorbidities diabetes and hypertension by ICD-9/10-CM codes or anti-diabetic and antihypertensive drug codes, and add anti-platelet and anti-lipid drugs into the original regression model. Model 3 was only tocompare the two cohorts (PPI vs. H2RA) in the original propensity-matched CKD cohort. Model 4 wasto add two comorbidities glomerulonephritis and acute tubular necrosis into propensity score matching and the original regression model listed in Table 2. In addition, we roughly addressed the association between acid suppression therapy and study outcomes in stages 1–4 vs. stage 5 CKD population by erythropoiesis-stimulating agent (ESA) and/or ICD-10-CM codes [22]. In Taiwan, ESA is only reimbursed to stage 5 CKD patients with anemia and can be used as a proxy for stage 5 CKD in prior NHI-based research [22,31]. CKD can be divided into five stages according to ICD-10-CM codes after 2016 [32], in addition to ESA (a proxy for stage 5 CKD). All data wereanalyzed using SAS (version 9.4; SAS Institute, Inc., Cary, NC, USA). A two-sided *p*-value less than 0.05 was considered statistically significant.

## 3. Results 

### 3.1. Demographic Data of the CKD Cohort

Before propensity matching, PPI, H2RA, and control cohorts accounted for 7%, 47%, and 46%, respectively, of the overall 102,802 CKD patients (Table 1). Compared to the H2RA and control cohorts, the PPI cohort was more likely to be male and older, have more medical visits, have a higher prevalence of comorbidities, and a more frequent use of NSAID and ACEI/ARB. A total of 13,083 matched CKD patients were obtained after 1:1:1 propensity matching; each cohort accounted for 4361 patients and did not differ in the above-mentioned matching variables. 

### 3.2. Association between Acid Suppression Therapy and Study Outcomes

Considering neither uses as the reference, the multivariable Cox regression revealed that the adjusted hazard ratios for ESRD in the presence of competing mortality were 0.4 (0.30–0.53, *p* < 0.0001) and 1.15 (0.91–1.45, *p* = 0.24) and that for overall mortality were 0.64 (0.57–0.72, *p* < 0.0001) and 1.83 (1.65–2.03, *p* < 0.0001) in the H2RA and PPI cohorts, respectively (Table 2). The association between H2RA and lower risks of ESRD and overall mortality remained despite considering PPI users as the reference (Appendix A). Further analysis of individual H2RA and PPI associated with study outcomes (Appendix A) demonstrated consistent results, with the exception of pantoprazole, which was associated with a significantly higher risk of ESRD. 

### 3.3. Cumulative Incidences of ESRD and Overall Mortality

During follow-up, 490 (3.7%) developed ESRD, 2572 (19.7%) died, and 2264 (17.3%) died before developing ESRD. Furthermore, 26.5% of the PPI cohort died before developing ESRD; this percentage was higher than the H2RA (13.2%) and control (12.2%) cohorts (*p* < 0.0001) (Table 3). The 15-year cumulative incidences of ESRD after considering death as a competing event (Appendix A) and overall mortality were significantly lower in the H2RA cohort (2.5%, 95% confidence interval: 2.0–3.2%; 31.5%, 27.7–35.3%), compared to the control (4.6%, 3.8–5.4%; 33.9%, 24.2–43.8%) and PPI cohorts (8.5%, 7.3–9.7%; 49.7%, 43.8–55.3%) (all *p* < 0.0001). When analyzing the Kaplan–Meier curve for ESRD-free survival in three study cohorts by log-rank test (Appendix A), survival seemed also to be more favorable for patients who took H2RAs, followed by those who took neither and those who took PPIs (*p* < 0.0001).

### 3.4. Dose–Response Relationship of Acid Suppression Therapywith ESRD, Overall, Cardiovascular, and Non-Cardiovascular Mortality

The association between the median cDDD and prescription frequency of H2RA and PPI and study outcomes was explored. Considering neither users as the reference, the H2RA cohort receiving ≥ 10 cDDD, compared with those receiving < 10 cDDD, showed significantly lowered risks across all the outcomes (Table 4). This dose-response relationship of H2RA with all study outcomes remained consistent when H2RA cDDD was divided into five levels of ≤5, 5–10, 10–15, 15–20, and >20 (Appendix A). However, there was no dose-response relationship of PPI by two levels of median dose or four levels (Appendix A) across all outcomes. More H2RA prescriptions were associated with significantly lower risks of ESRD and overall mortality, whereas there was no relationship between the frequency of PPI prescriptions and study outcomes (Table 5).

### 3.5. Sensitivity Analyses

The association between H2RA and lower risks of ESRD and overall mortality, as well as between PPI and higher overall mortality, seemed consistent in several sensitivity tests, including the exclusion of CKD patients who died or developed ESRD within 30, 60, and 90 days after the index date (Table 6), multivariable stratified analyses (Appendix A), and approaches using models 1 or 2 or 4 (Appendix A). 

### 3.6. Association between Acid Suppression Therapy and Study OutcomesRoughly by CKD Stages 1–4 vs. Stage 5

After propensity score matching, the associations between H2RA and lower ESRD and mortality risks and between PPI and higher mortality risk remained in stages 1–4 CKD populations(Appendix A). The association betweenacid suppression therapy and study outcomes in the stage 5 CKD population could not be addressed due to scanty numbers.

## 4. Discussion

To our knowledge, this is the first large nationwide cohort study of the CKD population using the new-user design, competing for mortality, and propensity score matching of PPI, H2RA, and neither substance to demonstrate a significantly lower risk for ESRD by 60% and overall mortality by 37% with H2RA use. These findings were reinforced by results from animal models suggesting plausible mechanisms for benefits [3,4,5]. Notably, these benefits of ESRD, overall, cardiovascular, and noncardiovascular mortality were dose-dependent and achieved more with higher cDDD of H2RA and more prescriptions. We also found a significantly higher risk of overall mortality by 83%, but not ESRD risk, with PPI use, and there was no dose-response relationship between PPI and all outcomes. Analyses of individual H2RA and PPI remained similar results. Our findings proved to be robust and consistent throughout multiple sensitivity analyses with careful control of confounders.

In disagreement with our results that H2RA exerts potential renal and survival benefits, a clinic-based non-matching cohort study of 25,455 CKD patients by Cholin et al. [21], which more closely resembled our three study cohorts and also took neither users as the reference, demonstrated no lower risks of ESRD and overall mortality in H2RA users. In the current study, we used propensity matching to minimize the baseline differences between three study cohorts, a new-user design to minimize immortal bias, and excluded patients with subsequent PPI or H2RA use during follow-up to eliminate interference and counteracting effects within each other. This may account for the discrepancy. Even so, further prospective cohort studies are warranted to corroborate this relationship of H2RA with the renal outcome. In agreement with our results that H2RA exerts potential heart and survival benefits, a Danish nationwide cohort study [14] of heart failure patientsreported lower risks of overall, cardiovascular, and non-cardiovascular mortality in H2RA than PPI users; a prospective cohort study [13] of participants without cardiovascular disease at baseline demonstrated a 62% reduced risk of new-onset heart failure in H2RA users; a meta-analysis study [15] documented improved heart function in H2RA users; two retrospective, propensity-matched observational studies [16,17] reported improved survival in hospitalized patients with COVID-19 treated with famotidine. Takentogether, these results concurred with previous observations in animal models to suggest a biologically plausible histamine-related mechanism important to the pathogenesis of renal injury [3,4], cardiovascular health [5,13,14,15,16], and sepsis [33]. 

Experimental evidence has accumulated over time, suggesting the pathophysiological roles of histamine and H2R that belong to theG-protein-coupled-receptor family [3,4]. In response to histamine, H2R activation leads to adenylyl cyclase-mediated cAMP activation, which can stimulate renin release in the kidney, induces positive inotropic and chronotropic responses of the heart [3,4,5], and this closely relates to the development of various cardiovascular diseases, such as myocardial ischemia and infarction, hypertension, and heart failure [15]. High plasma histamine levels have been found in patients with renal insufficiency and ESRD, which is consistent with histamine’s ability to reduce urea clearance [3,4]. Histamine is also a well-known mediator of inflammation [4], which isintimately linked to acute and chronic renal diseases [34] and also interacts between renal parenchymal cells and resident immune cells, such as macrophagesand neutrophils [35]. H2R has also been implicated in the immune response [4]. In response to stimulus-evoked inflammation, mast cells release proinflammatory cytokines and vasoactive histamine; histamine synthesis can also be induced by macrophages and neutrophils [2,4]. Histamine can modulate activated neutrophil oxidative burst linked to the production of reactive oxygen species via H2R, which is also expressed in neutrophils [2]. Therefore, it is the vasoactive and inflammation-related functions of histamine that have the greatest relevance to renal function [4]. Further H2RA interventional animal studies have helped strengthen the evidence on the detrimental roles of histamine and H2R. In animal models of ischemic acute renal failure, cimetidine improved renal function [36,37]; meanwhile, ranitidine reduced renal damage, attenuated atherosclerosis, and increased survival following renal ischemia [38]. In animal models of sepsis, which demonstrated a several-foldincrease in plasma histamine, famotidine reduced blood and tissue levels of interleukin-6,interleukin-1β, tumor necrosis factor-α, and their mRNAs, and decreased blood urea nitrogen and serum creatinine [33], whereascimetidine lowered mortality [39]. In animal models of pressure-induced heart failure, famotidine improved cardiac function and reduced cardiac hypertrophy and fibrosis [5]. In studies on human heart samples, and cimetidine and ranitidine antagonized the inotropic and chronotropic effects [5]. Given increased plasma histamine levels and inflammatory response in CKD patients and the presence of H2R in both kidney and heart, our results provide clinical evidence linking H2RA to reno- and cardioprotection. 

It is note worthy that this is the first human report on the dose-dependent effect of H2RA on kidney, heart, and survival outcomes, which builds on prior experimental research, after adjusting for the number of medical visits. In an experimental study, the effect of histamine on renal vascular resistance via H2R was dose-dependent; a higher dose of histamine caused a drop in creatinine and urea clearance and stimulated renal sympathetic nerve activity [3,4]. This appeared to account for our results that a higher dose of H2RA achieved greater renal benefit. In vivo studies on human neutrophil function revealed that both cimetidine and famotidine inhibited reactive oxygen species production of neutrophilsin a dose-dependent manner [2]. Remarkably, high-dose oral famotidine was associated with symptomatic improvement in non-hospitalized patients with COVID-19 [18]. Further prospective research is warranted to better understand the causal relationship and pathological mechanism underlying this association. 

We found the statistically non-significant relationship of PPI with ESRD risk in fully multivariable regression, indicating that the adverse renal effect of PPI was dwarfed by the complex covariate influences that might lead to renal injury. We also found the excess risk of death associated with PPI use in CKD patients was not in a dose-dependent manner. Some discordant associations of PPI with renal and survival outcomes existed in prior research onrenal transplants, CKD patients, and normal renal function populations. When considering H2RA users as the reference in the normal renal function population, one study reported no significantly increased risks for AKI and ESRD in PPI users [7]; three studies reported increased risks for AKI [6], CKD [6,8,9], ESRD [8,9], and death [10,11] in PPI users. When considering non-PPI users as the reference, one study of renal transplants [40] reported increased risk for death in PPI users, especially high dose, but not graft failure, and estimated glomerular filtration rate decline; a study of CKD patients [41] reported increased risk for major adverse renal events in PPI users, but not death. When considering neither user of H2RA or PPI as the reference, a study of CKD patients [21] reported no increased risks for ESRD and death in PPI and H2RA users. The matching method and choice of control reference seemed to accountfor this discrepancy. Moreover, the lack of animal and interventional evidence elucidating biological plausibility that linked PPI to renal and cardiovascular complications [9,12] critically weakened the importance of this relationship. PPI-induced acute interstitial nephritis and acute kidney injury have proven causality, and the most reasonable pathophysiological mechanism seems to be an idiosyncratic reaction [12]. However, the association of CKD, cardiovascular events, and pneumonia with PPI is of low magnitude and insufficient evidence for causality; even the incidence of CKD was not of clinical concern [12]. The following two presumed mechanisms based on in vitro experimental data of PPI on vascular homeostasis have been proposed: a direct interference with the endothelial homeostasis of asymmetric dimethylarginine (ADMA), an endogenous inhibitor of nitric oxide synthase, and the acceleration of endothelial senescence [12]. Plasma ADMA increases with the severity of CKD and might contribute to the increased cardiovascular mortality in these patients [42], which may also account, in part, for our findings. However, another trial conducted among healthy subjects and coronary disease patients reported no significant influence onvascular endothelial function [43]. Another in vitro effect of PPIs on endothelial cell aging was not ready for translation into a clinical setting [12]. Future prospective research is warranted to mechanistically explain the correlation between PPI use and adverse clinical outcomes.

By analyzing the NHI claims data with a highly representative sample, the present study demonstrated five strengths. First, recall bias of acid-suppression agents was avoided, and cumulative dose and prescription frequency could be identified. Second, the new-use design minimized the immortal bias and the potential residual effect of using acid-suppression agents before CKD diagnosis. Third, follow-up of ESRD and death outcomes was completed, and theuse of competing mortality minimized the risk of overestimating the ESRD. Fourth, the use of propensity score matching and stratified analyses minimized confounding effects. Fifth, consideration of the use of medical services minimized detection bias [23,24,25]. However, several potential limitations exist. First, the compliance of prescribed acid-suppression agents was not assessed in the administrative claims. Second, some patients may have usedself-paid acid suppressants and thus may have been inappropriately classified into the control cohort. These potential misclassifications may have led to an overestimation or underestimation of the associations in the study. Third, the NHI claims data lack information on family history, the primary cause of kidney diseases, lifestyle (e.g., smoking, alcohol consumption, diet, and physical activity), body weight, blood pressure and sugar results, and other laboratory data (e.g., creatinine and exact CKD stage), which may contribute to the risks of ESRD and death. Fourth, unmeasured confounders may still exist as in any observational study. Fifth, a causal association between acid suppressants and study outcomes cannot be inferred based on an observational study. Finally, the exact causes and underlying pathologies that induced CKD could not be assessed in Taiwan’s NHI datasets.

## 5. Conclusions

This national cohort study on CKD patients indicated that H2RA, especially in a dose-response relationship, was associated with areduced risk of ESRD, overall, cardiovascular, and non-cardiovascular mortality, while PPI, not in a dose-response relationship, was associated with anincreased risk of overall mortality, but not ESRD. Further prospective research is warranted to improve our understanding of the causal and dose-response relationships underlying this association.

## Figures and Tables

**Figure 1 jcm-11-05612-f001:**
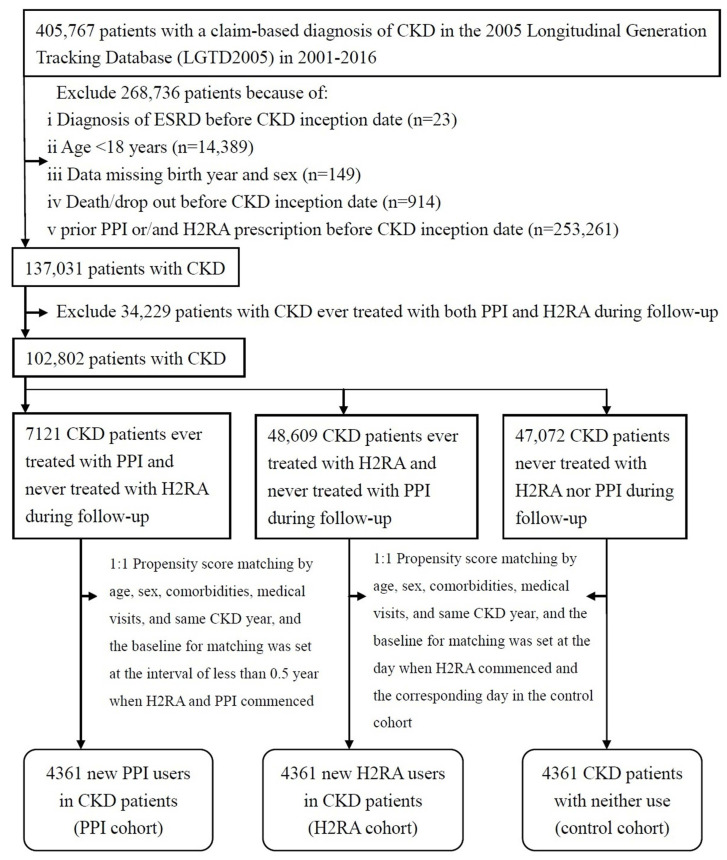
Flow diagram of the enrollment process. Abbreviations: CKD, chronic kidney disease; ESRD, end-stage renal disease; H2RA, H2-receptor antagonist; PPI, proton pump inhibitor.

**Table 1 jcm-11-05612-t001:** Baseline characteristics of the three study cohorts in the CKD patients.

	Overall CKD Patients (*n* = 102,802)				Propensity-Matched CKD Patients (*n* = 13,083)			
	PPI Cohort	H2RA Cohort	Control		PPI Cohort	H2RA Cohort	Control	
Variables	(*n* = 7121) N(%)	(*n* = 48,609) N(%)	(*n* = 47,072) N(%)	*p*-Value	(*n* = 4361) N (%)	(*n* = 4361) N (%)	(*n* = 4361) N (%)	*p*-Value
Sex				<0.0001				0.99
Men	4524 (63.5)	23,825 (49.0)	27,931 (59.3)		2831 (64.9)	2831 (64.9)	2836 (65.0)	
Women	2597 (36.5)	24,784 (51.0)	19,141 (40.7)		1530 (35.1)	1530 (35.1)	1525 (35.0)	
Age (year)				<0.0001				1.00
18–45	1610 (22.6)	21,029 (43.3)	18,472 (39.2)		1391(31.9)	1386 (31.8)	1390 (31.9)	
46–55	1325 (18.6)	10,373 (21.3)	10,075 (21.4)		959 (22.0)	958 (22.0)	954 (21.9)	
56–65	1334 (18.7)	7796 (16.1)	8398 (17.8)		804 (18.4)	814 (18.7)	811 (18.6)	
66–75	1381 (19.4)	5944 (12.2)	5631 (12.0)		670 (15.4)	665 (15.2)	666 (15.2)	
>75	1471 (20.7)	3467 (7.1)	4496 (9.6)		537 (12.3)	538 (12.3)	540 (12.4)	
Mean (SD)	58.8 ± 17.3	48.3 ± 17.2	50.4 ± 17.3	<0.0001	53.7 ± 17.2	52.8 ± 17.8	53.0 ± 17.5	0.03
Comorbidities								
Diabetes	1825 (25.6)	6969 (14.3)	9387 (19.9)	<0.0001	584 (13.4)	579 (13.3)	578 (13.3)	0.98
Coronary heart disease	1076 (15.1)	3926 (8.1)	3409 (7.2)	<0.0001	200 (4.6)	209 (4.8)	200 (4.6)	0.87
Hypertension	2779 (39.0)	11,778 (24.3)	12,821 (27.2)	<0.0001	1148 (26.3)	1145 (26.3)	1143 (26.2)	0.99
Acid peptic disease	555 (7.8)	3182 (6.6)	1957 (4.1)	<0.0001	86 (2.0)	93 (2.1)	82 (1.9)	0.69
Chronic liver disease	841 (11.8)	5188 (10.7)	4114 (8.7)	<0.0001	220 (5.0)	218 (5.0)	224 (5.1)	0.96
No. of medical visits				<0.0001				0.99
1–11	3067 (43.1)	22,183 (45.7)	27,460 (58.4)		2389 (54.8)	2379 (54.6)	2390 (54.8)	
12–23	2346 (32.9)	16,150 (33.2)	13,295 (28.2)		1288 (29.5)	1295 (29.7)	1297 (29.7)	
>23	1708 (24.0)	10,276 (21.1)	6317 (13.4)		684 (15.7)	687 (15.7)	674 (15.5)	
Mean (SD)	17.0 ± 14.5	16.0 ± 13.2	12.7 ± 11.6	<0.0001	13.6 ± 12.6	13.9 ± 12.6	13.6 ± 12.9	0.30
Confounding drugs								
NSAID	2620 (36.8)	11,756 (24.2)	1220 (2.6)	<0.0001	1362 (31.2)	1209 (27.7)	85 (2.0)	<0.0001
ACEI/ARB	2149 (30.2)	5542 (11.4)	1251 (2.7)	<0.0001	920 (21.1)	558 (12.8)	83 (1.9)	<0.0001

Abbreviations: ACEI/ARB, angiotensin-converting enzyme inhibitor/angiotensin receptor blocker; CKD, chronic kidney disease; H2RA, H2-receptor antagonist; NSAID, nonsteroid anti-inflammatory drug; PPI, proton pump inhibitor; SD, standard deviation.

**Table 2 jcm-11-05612-t002:** Hazard ratios (HRs) for end-stage renal disease (ESRD) and overall mortality in three cohorts.

Outcome	Crude			Adjusted		
	HR	95% CI	*p*-Value	HR	95% CI	*p*-Value
ESRD *						
Control (*n* = 4361)	1.00	Reference		1.00	Reference	
PPI cohort (*n* = 4361)	2.02	1.65–2.48	<0.0001	1.15	0.91–1.45	0.24
H2RA cohort (*n* = 4361)	0.57	0.43–0.75	<0.0001	0.40	0.30–0.53	<0.0001
Overall mortality ^#^						
Control (*n* = 4361)	1.00	Reference		1.00	Reference	
PPI cohort (*n* = 4361)	2.54	2.31–2.80	<0.0001	1.83	1.65–2.03	<0.0001
H2RA cohort (*n* = 4361)	0.98	0.88–1.09	0.70	0.64	0.57–0.72	<0.0001

Abbreviations: CI, confidence interval; others same as Table 1. * Adjusted for all covariates (age per year, sex, comorbidities, number of medical visits, NSAID, and ACEI/ARB) and competing mortality. ^#^ Adjusted for all covariates (age per year, sex, comorbidities, number of medical visits, NSAID, and ACEI/ARB).

**Table 3 jcm-11-05612-t003:** End-stage renal disease (ESRD) occurrence and overall mortality over a 15-year follow-up period.

	PPI Cohort (*n* = 4361)	H2RA Cohort (*n* = 4361)	Control (*n* = 4361)	*p*-Value
ESRD				
Follow-up (years), mean ± SD	3.5 ± 3.5	4.8 ± 3.7	4.5 ± 3.6	
Event number, *n* (%)	270 (6.2)	81 (1.9)	139 (3.2)	<0.0001
Competing mortality, *n* (%)	1155 (26.5)	575 (13.2)	534 (12.2)	<0.0001
Cumulative incidence (%)	8.5 (95% CI, 7.3–9.7)	2.5 (95% CI, 2.0–3.2)	4.6 (95% CI, 3.8–5.4)	<0.0001
Overall mortality				
Follow-up (years), mean ± SD	3.5 ± 3.5	4.8 ± 3.7	4.5 ± 3.6	
Event number, *n* (%)	1315 (30.2)	631 (14.5)	626 (14.4)	<0.0001
Cumulative incidence (%)	49.7 (95% CI, 43.8–55.3)	31.5 (95% CI, 27.7–35.3)	33.9 (95% CI, 24.2–43.8)	<0.0001

Abbreviations: the same as Table 1 and Table 2.

**Table 4 jcm-11-05612-t004:** Association between median cumulative define daily dose (cDDD) of acid suppressants and study outcomes.

	ESRD			Overall Mortality			CV Mortality			Non-CV Mortality		
Taking Controls as the Reference	Events (%)	aHR * (95% CI)	*p*-Value	Events (%)	aHR ^#^ (95% CI)	*p*-Value	Events (%)	aHR ^#^ (95% CI)	*p*-Value	Events (%)	aHR ^#^ (95% CI)	*p*-Value
PPI cDDD												
<30 (*n* = 2198)	157 (7.1)	1.18 (0.86–1.62)	0.32	889 (40.4)	2.20 (1.92–2.54)	<0.0001	115 (5.2)	2.01 (1.33–3.06)	0.001	774 (35.2)	2.22 (1.91–2.58)	<0.0001
≥30 (*n* = 2163)	113 (5.2)	1.24 (0.87–1.76)	0.23	426 (19.7)	1.37 (1.16–1.62)	0.0003	42 (1.9)	1.30 (0.77–2.18)	0.33	384 (17.8)	1.38 (1.15–1.65)	0.0005
H2RA cDDD												
<10 (*n* = 2326)	50 (2.1)	0.45 (0.30–0.68)	0.0001	335 (14.4)	1.04 (0.88–1.23)	0.65	65 (2.8)	1.86 (1.14–3.02)	0.012	270 (11.6)	0.95 (0.79–1.14)	0.59
≥10 (*n* = 2035)	31 (1.5)	0.32 (0.20–0.51)	<0.0001	296 (14.5)	0.50 (0.42–0.60)	<0.0001	29 (1.4)	0.45 (0.27–0.75)	0.002	267 (13.1)	0.51 (0.43–0.61)	<0.0001

Abbreviations: aHR, adjusted hazard ratio; CV, cardiovascular; others same as Table 1 and Table 2. * Adjusted for all covariates (age per year, sex, comorbidities, number of medical visits, NSAID, and ACEI/ARB) and competing mortality. ^#^ Adjusted for all covariates (age per year, sex, comorbidities, number of medical visits, NSAID, and ACEI/ARB).

**Table 5 jcm-11-05612-t005:** Association between the frequency of acid-suppressant prescription and study outcomes.

Frequency of Prescriptions	ESRD			Overall Mortality		
	Event	aHR * (95%CI)	*p*-Value	Event	aHR ^#^ (95%CI)	*p*-Value
PPI						
0 (*n* = 4361)	139	1 (reference)		626	1 (reference)	
1 (*n* = 1922)	125	1.22 (0.92–0.62)	0.17	687	2.51 (2.22–2.83)	<0.0001
2 (*n* = 798)	41	1.11 (0.78–1.60)	0.56	197	1.61 (1.36–1.90)	<0.0001
3–4 (*n* = 838)	36	0.92 (0.63–1.35)	0.68	170	1.34 (1.13–1.60)	0.001
≥5 (*n* = 808)	68	1.46 (1.07–1.99)	0.016	261	1.34 (1.15–1.56)	0.0001
H2RA						
0 (*n* = 4361)	139	1 (reference)		626	1 (reference)	
1 (*n* = 1595)	38	0.52 (0.36–0.77)	0.0011	258	1.25 (1.07–1.47)	0.004
2 (*n* = 742)	8	0.24 (0.12–0.50)	0.0001	93	0.77 (0.62–0.96)	0.02
3–4 (*n* = 771)	18	0.47 (0.28–0.79)	0.004	91	0.56 (0.45–0.70)	<0.0001
≥5 (*n* = 1253)	17	0.28 (0.16–0.46)	<0.0001	189	0.44 (0.37–0.52)	<0.0001

Abbreviations: the same as Table 1, Table 2, Table 3 and Table 4. * Adjusted for all covariates (age per year, sex, comorbidities, number of medical visits, NSAID, and ACEI/ARB) and competing mortality. ^#^ Adjusted for all covariates (age per year, sex, comorbidities, number of medical visits, NSAID, and ACEI/ARB).

**Table 6 jcm-11-05612-t006:** Risks of study outcomes in CKD patients excluding dying or developing ESRD within 30, 60, and 90 day after the index date.

			ESRD			Overall Mortality	
		N	Crude HR (95% CI)	Adjusted HR * (95% CI)	N	Crude HR (95% CI)	Adjusted HR ^#^ (95% CI)
Follow-up >30 days	Control	4288	1.00 (Reference)	1.00 (Reference)	4292	1.00 (Reference)	1.00 (Reference)
	PPI	3873	1.69 (1.35–2.11)	1.14 (0.89–1.47)	3914	1.95 (1.76–2.17)	1.66 (1.48–1.85)
	H2RA	4227	0.54 (0.40–0.72)	0.40 (0.30–0.54)	4235	0.87 (0.78–0.98)	0.62 (0.55–0.70)
Follow-up >60 days	Control	4217	1.00 (Reference)	1.00 (Reference)	4226	1.00 (Reference)	1.00 (Reference)
	PPI	3664	1.53 (1.21–1.93)	1.11 (0.85–1.44)	3717	1.73 (1.56–1.93)	1.57 (1.40–1.76)
	H2RA	4166	0.51 (0.38–0.69)	0.39 (0.28–0.53)	4176	0.86 (0.76–0.97)	0.64 (0.57–0.73)
Follow-up >90 days	Control	4132	1.00 (Reference)	1.00 (Reference)	4141	1.00 (Reference)	1.00 (Reference)
	PPI	3525	1.39 (1.09–1.78)	1.06 (0.81–1.38)	3586	1.66 (1.48–1.85)	1.56 (1.39–1.76)
	H2RA	4090	0.50 (0.36–0.67)	0.38 (0.28–0.52)	4101	0.85 (0.76–0.96)	0.65 (0.57–0.74)

Abbreviations: the same as Table 1 and Table 2. * Adjusted for all covariates (age per year, sex, comorbidities, number of medical visits, NSAID, and ACEI/ARB) and competing mortality. ^#^ Adjusted for all covariates (age per year, sex, comorbidities, number of medical visits, NSAID, and ACEI/ARB).

## Data Availability

Restrictions apply to the availability of these data. Data were obtainedfrom National Health Insurance database and are available from the authors with the permission of National Health Insurance Administration of Taiwan.

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
