# Peer review of "Impact of Acid Suppression Therapy on Renal and Survival Outcomes in Patients with Chronic Kidney Disease: A Taiwanese Nationwide Cohort Study"

_jcm, 2022, doi:10.3390/jcm11195612_

Round 1

Reviewer 1 Report

jcm-1872732

 The authors evaluated the effects of histamine-2-receptor antagonist (H2RA) and proton pump inhibitor (PPI) on renal and survival outcomes in chronic kidney disease (CKD) patients using Taiwanese national health insurance database. They found H2RA were associated with reduced risk of end-stage renal disease (ESRD) in dose-response relationship. Some major issues need to be addressed.

 Major comments:

1. The critical weak point of the study is lack of renal function validation in each group CKD subjects. The author should analyze CKD stage by ICD coding even lack information of laboratory data. Following, they should compare the impacts of H2RA and PPI on renal and survival outcomes at different stage of CKD. More precise information can be confirmed and provided by this study.

2. Kaplan-Meier curve to present the cumulative incidence of ESRD in each cohort were suggested to be provided. It is better if providing subgroup analyses of different stages of CKD.

3. We suggest to provide subgroup analyses depicting the hazard ratio (HR) and 95% CI for the risk of ESRD (or mortality) associated with H2RA use.

4. There were specific indications for use of H2RA or PPI. Whether the underline disease indicated for these two drugs use all the same? PPIs were prescribed for more severe GI disease and H2RA were prescribed for less severe subjects. If yes, it is reasonable to compare H2RA with PPI. On the other hand, whether we ignore the impacts of underline disease when comparing with control group with drugs used groups?

5. The more frequent visit and larger H2RA dosage prescribed, the more renoprotective effects? More discussion, proposed mechanisms, and evidence should be provided to convince the summary.

Author Response

Sep 11, 2022

JCM Editorial Office

Journal of Clinical Medicine

Re: jcm-1872732

Dear Editors,

Thanks for your letter on Aug 26, 2022 regarding our article titled as:  Impact of Acid-Suppression Therapy on Renal and Survival Outcomes in Patients with Chronic Kidney Disease: A Taiwanese Nationwide Cohort Study. We've learned a lot from your valuable advice. According to your suggestion, we make some revisions marked as red words in the revised manuscript. More clear details are described as the follows.

Response to Reviewer 1

  1. The critical weak point of the study is lack of renal function validation in each group CKD subjects. The author should analyze CKD stage by ICD coding even lack information of laboratory data. Following, they should compare the impacts of H2RA and PPI on renal and survival outcomes at different stage of CKD. More precise information can be confirmed and provided by this study.

Response: The Taiwan NHI's datasets lack information on laboratory data (eg. creatinine) and uses ICD-9-CM (before 2016) and ICD-10-CM (after 2016) diagnosis codes to define diseases (Ref 32). Thus, the exact CKD stage could not be assessed in the current LGTD2005 and listed it as a limitation (page 18, lines 7-8). Before 2016, CKD could be roughly divided into two groups by erythropoiesis-stimulating agent (ESA): stages 1-4 and stage 5. In Taiwan, ESA is only reimbursed to stage 5 CKD patients with anemia and can be used as a proxy for stage 5 CKD in prior NHI-based research (Ref 22, 31). After 2016, CKD could be divided into five stages according to ICD-10-CM codes, in addition to ESA (a proxy for stage 5 CKD). Thus, we can roughly divide our CKD population into two groups: CKD stages 1-4 vs. stage 5 by ESA and/or ICD-10-CM codes. After propensity score matching, the significant association of H2RA with lower ESRD and death risks and that of PPI with higher death risk remained consistent in stages 1-4 CKD population. However, the association remained unresolved in stage 5 CKD population due to limited numbers. We added these results in the supplementary Table S5, added three references (Ref 22, 31, 32), and revised our manuscript (page 9, lines 20-22; page 10, lines 1-3; page 12, lines 16-21).

  1. Kaplan-Meier curve to present the cumulative incidence of ESRD in each cohort were suggested to be provided. It is better if providing subgroup analyses of different stages of CKD.

Response: To address the reviewer's concern, we provided Kaplan-Meier curve for ESRD-free survival and cumulative incidence of ESRD after adjustment for competing mortality using modified Kaplan–Meier and Grey methods among H2RA, PPI, and control cohorts (Supplementary Figure S1). Both results remained consistent. the exact CKD stage could not be assessed in the current LGTD2005 and listed it as a limitation (page 18, lines 7-8). We revised our manuscript (page 8, lines 17-18; page 11, lines 9-10, 13-16).

  1. We suggest to provide subgroup analyses depicting the hazard ratio (HR) and 95% CI for the risk of ESRD (or mortality) associated with H2RA use.

Response: To address the reviewer's concern, we provided forest plot of ESRD and overall mortality associated with H2RA use (vs. nonuse) in the Supplementary Figure S2. We revised our manuscript (page 12, lines 13-14).

  1. There were specific indications for use of H2RA or PPI. Whether the underline disease indicated for these two drugs use all the same? PPIs were prescribed for more severe GI disease and H2RA were prescribed for less severe subjects. If yes, it is reasonable to compare H2RA with PPI. On the other hand, whether we ignore the impacts of underline disease when comparing with control group with drugs used groups?

Response: CKD is associated with increased incidence of acid peptic disease (Ref 19), which can be treated with antacids (Ref 20) containing ingredients such as aluminum, calcium, magnesium, or sodium bicarbonate, H2RAs, and PPIs. Table 1 shows the percentage of acid peptic disease in PPI, H2RA, and control cohorts was 7.8%, 6.6%, and 4.1%, respectively, before propensity score matching. In Taiwan, antacids and H2RA can be prescribed for patients having symptoms of acid peptic disease without endoscopic examinations; however, PPIs, which are costly, are only reimbursed for patients who complete an endoscopic examination showing any type of acid peptic disease and prescribed only for 4 months since inspection date of each endoscopy. In our study, CKD patients without the use of PPIs or H2RAs during follow-up were counted as nonusers (i.e., controls), which was also used in another study (Ref 21) of CKD patients investigating the association between PPI, H2RA, and renal and survival outcomes. Due to literature does not demonstrate that antacids may increase or decrease ESRD risk, we did not exclude antacids from three study cohorts. Thus, neither users of H2RA or PPI (as controls) may or may not use antacids. To address the reviewer's concern, we revised our manuscript (page 5, lines 6-12, 15; page 6, line 22; page 7, line 1).

  1. The more frequent visit and larger H2RA dosage prescribed, the more renoprotective effects? More discussion, proposed mechanisms, and evidence should be provided to convince the summary.

Response: We included the number of medical visits in the propensity score matching and regression model to minimize the detection bias (Ref 23, 24) because of easy accessibility and availability of medical services in Taiwan and confounding effect of medical attention as medical attention may explain some of the remaining risk elevation (Ref 23). After propensity matching, there was no difference in the number of medical visits in three cohorts (Table 1). Nevertheless, H2RA remained renoprotective. To address the reviewer's concern, we further divided H2RA cDDD to five levels (5, 10, 15, 20, and >20) to delineate the association of H2RA with study outcomes (Supplementary Table S6). We also found the dose-response relationship of H2RA with study outcomes, except in the levels of 10<cDDD≤15 and 15<cDDD≤20 due to fewer cases. The current study is the first human report on dose-dependent effect of H2RA on kidney, heart, and survival outcomes, which build on prior experimental research, after adjusting the number of medical visits. More discussion, proposed mechanisms, and experimental evidence were shown in the 3rd and 4th paragraphs of discussion section (pages 14-16). We revised our manuscript (page 9, lines 5-6; page 12, lines 1-3).

Thank you heartily for your invaluable opinions on this paper. We are deeply honored by the time and efforts that you had spent in reviewing and revising this manuscript. By incessantly reviewing and revising our texts, we are spurred to read more and learn more from your comments.

Sincerely Yours,

Yi-Chun Chen, MD

Division of Nephrology, Department of Internal Medicine, Buddhist Dalin Tzu Chi General Hospital, Chiayi, and School of Medicine, Tzu Chi University, Hualien, Taiwan

Reviewer 2 Report

The article "Impact of Acid-Suppression Therapy on Renal and Survival Outcomes in Patients with Chronic Kidney Disease: A Taiwanese Nationwide Cohort Study" by Yi-Chun Chen, Yen-Chun Chen, Wen-Yen Chiou and Ben-Hui is presented for peer review. In this article, the authors (experts in the field) discuss the effect of the H2RA dose on ESRD risk reduction and overall mortality in patients with CKD. While the article is thoroughly and well written, it can be further improved by enhancing its readability.

1.      Correct typos in the manuscript (e.g. line 15).

2.      Please standardise the thousands separator throughout the manuscript. I suggest using space, the internationally recommended thousands separator.

3.      Figures and tables. Provide abbreviations in alphabetical order.

'H2-receptor antagonist' - remove underlining.

4.      Table 3 is before table 2- please correct.

5.      Table 5. line 200. Sign "≧"?

6.      References. Remove the underlining of words.

Author Response

Sep 11, 2022

JCM Editorial Office

Journal of Clinical Medicine

Re: jcm-1872732

Dear Editors,

Thanks for your letter on Aug 26, 2022 regarding our article titled as:  Impact of Acid-Suppression Therapy on Renal and Survival Outcomes in Patients with Chronic Kidney Disease: A Taiwanese Nationwide Cohort Study. We've learned a lot from your valuable advice. According to your suggestion, we make some revisions marked as red words in the revised manuscript. More clear details are described as the follows.

Response to Reviewer 2

  1. Correct typos in the manuscript (e.g. line 15).

Response: To address the reviewer's concern, we corrected typos throughout the manuscript.

  1. Please standardise the thousands separator throughout the manuscript. I suggest using space, the internationally recommended thousands separator.

Response: To address the reviewer's concern, we standardise the thousands separator with space throughout the manuscript and Tables.

  1. Figures and tables. Provide abbreviations in alphabetical order. 'H2-receptor antagonist' - remove underlining.

Response: To address the reviewer's concern, we revised abbreviations in alphabetical order in all Tables and removed the underlining of H2-receptor antagonist.

  1. Table 3 is before table 2- please correct.

Response: To address the reviewer's concern, we changed "Table 3" to "Table 2" and "Table 2" to "Table 3". We revised our manuscript (page 10, lines 16, 20; page 11, lines 5, 9).

  1. Table 5. line 200. Sign "≧"?

Response: To address the reviewer's concern, we corrected sign "≧" to "≥" in the revised Table 5.

Thank you heartily for your invaluable opinions on this paper. We are deeply honored by the time and efforts that you had spent in reviewing and revising this manuscript. By incessantly reviewing and revising our texts, we are spurred to read more and learn more from your comments.

Sincerely Yours,

Yi-Chun Chen, MD

Division of Nephrology, Department of Internal Medicine, Buddhist Dalin Tzu Chi General Hospital, Chiayi, and School of Medicine, Tzu Chi University, Hualien, Taiwan

Reviewer 3 Report

The manuscript Impact of Acid-Suppression Therapy on Renal and Survival 2 Outcomes in Patients with Chronic Kidney Disease: A Taiwan- 3ese Nationwide Cohort Study has an interesting, actual topic, with a clear and easy to be individualized title on literature research. The abstract is well structured and the data are clearly presented.

The Introduction is well documented, but I recommend some more information about the role of PPI and H2RA in renal diseases should be included.
Material and Methods are well conceived, to make the study reproducible, but I consider that the results are not concluding as long as the group of patients with CKD is not analyzed regarding the underling pathologies that induced CKD. I recommend a Table with the underling diseases and a short analyze between the tubular and glomerular diseases that lead to CKD in relation with PPI and H2RA use.

The Results should include more data about the cause of renal pathology and the relation of the cause with PPI and H2RA use.
Discussions are too long and difficult to read. I recommend a special paragraph of the studies that have similar results and the studies wih discordant results.
The Conclusions reflect the idea of the title and the manuscript presents a recent bibliography, with a reasonable number of titles.

Author Response

Sep 11, 2022

JCM Editorial Office

Journal of Clinical Medicine

Re: jcm-1872732

Dear Editors,

Thanks for your letter on Aug 26, 2022 regarding our article titled as:  Impact of Acid-Suppression Therapy on Renal and Survival Outcomes in Patients with Chronic Kidney Disease: A Taiwanese Nationwide Cohort Study. We've learned a lot from your valuable advice. According to your suggestion, we make some revisions marked as red words in the revised manuscript. More clear details are described as the follows.

Response to Reviewer 3

  1. The Introduction is well documented, but I recommend some more information about the role of PPI and H2RA in renal diseases should be included.

Response: To address the reviewer's concern, we revised our manuscript (page 4, lines 15-16, 21-22; page 5, line 1).

  1. (1) Material and Methods are well conceived, to make the study reproducible, but I consider that the results are not concluding as long as the group of patients with CKD is not analyzed regarding the underling pathologies that induced CKD. (2) I recommend a Table with the underling diseases and a short analyze between the tubular and glomerular diseases that lead to CKD in relation with PPI and H2RA use.

Response: (1) The exact underling pathologies that induced CKD could not be assessed in the Taiwan's NHI datasets, which was listed as a limitation (page 18, lines 11-12). However, such comorbidities as diabetes, hypertension, coronary heart disease, and chronic liver disease were also common reasons that induced CKD. Multivariable stratified analyses for the association between acid-suppression therapy and study outcomes were shown in Supplementary Table S2 and Supplementary Figure S2.

(2) We added two comorbidities glomerulonephritis and acute tubular necrosis into propensity score matching and the original regression model listed in Table 2. The significant association of H2RA with lower ESRD and death risks and that of PPI with higher death risk remained consistent. We listed the result as a sensitivity analysis in the revised supplementary Table S3 and revised our manuscript (page 9, lines 13, 18-20).

  1. The Results should include more data about the cause of renal pathology and the relation of the cause with PPI and H2RA use.

Response: The exact causes and underlying pathologies that induced CKD could not be assessed in the Taiwan's NHI datasets, which was listed as a limitation (page 18, lines 11-12).

  1. Discussions are too long and difficult to read. I recommend a special paragraph of the studies that have similar results and the studies with discordant results.

Response: To address the reviewer's concern, we shortened our discussion, presented similar results and the studies with discordant results (pages 13-14, 16-17), and revised all the discussion in the revised manuscript (page 13-18).

Thank you heartily for your invaluable opinions on this paper. We are deeply honored by the time and efforts that you had spent in reviewing and revising this manuscript. By incessantly reviewing and revising our texts, we are spurred to read more and learn more from your comments.

Sincerely Yours,

Yi-Chun Chen, MD

Division of Nephrology, Department of Internal Medicine, Buddhist Dalin Tzu Chi General Hospital, Chiayi, and School of Medicine, Tzu Chi University, Hualien, Taiwan

Round 2

Reviewer 1 Report

All the issues raised by me had been well answered.  It is now suitable for publication.

Reviewer 3 Report

The revised form of the manuscript respects the reviewers comments. I recommend acceptance in present form.